# The Protective Effects of Corn Oligopeptides on Acute Alcoholic Liver Disease by Inhibiting the Activation of Kupffer Cells NF-κB/AMPK Signal Pathway

**DOI:** 10.3390/nu14194194

**Published:** 2022-10-08

**Authors:** Ying Wei, Mingliang Li, Zhiyuan Feng, Di Zhang, Meiling Sun, Yong Wang, Xiangning Chen

**Affiliations:** 1Department of Food Science and Engineering, Beijing University of Agriculture, Beijing 102206, China; 2School of Food Science and Technology, Jiangnan University, Wuxi 214122, China; 3Key Laboratory of Industrial Fermentation Microbiology, Ministry of Education, Tianjin University of Science and Technology, Tianjin 300457, China; 4School of Food and Biological Engineering, Jiangsu University, Zhenjiang 212013, China; 5Academy of National Food and Strategic Reserves Administration, Beijing 100037, China

**Keywords:** corn oligopeptide, ALD, Kupffer cell, inflammatory cytokines, NF-κB/AMPK

## Abstract

Alcohol can cause injury and lead to an inflammatory response in the liver. The NF-κB/AMPK signaling pathway plays a vital role in regulating intracellular inflammatory cytokine levels. In this study, corn oligopeptides (CPs), as the research objects, were obtained from corn gluten meal, and their regulation of the activation of the Kupffer cell NF-κB/AMPK signal pathway induced by LPS was investigated. Results showed that ALT, AST, and inflammatory cytokines in mice serum after the administration of CPs at 0.2, 0.4, and 0.8 g/kg of body weight displayed a distinct (*p* < 0.05) reduction. On the other hand, the CPs also inhibited the expression of recognized receptor CD14 and TLR4, down-regulated P-JNK, P-ERK, and P-p-38, and thus inhibited inflammatory cytokine levels in Kupffer cells (KCs). Furthermore, four kinds of dipeptides with a leucine residue at the C-terminus that might exhibit down-regulated inflammatory cytokines in the NF-κB/AMPK signaling pathway functions were detected using HPLC-MS/MS. These results indicated that CPs have a potential application value in acute alcoholic liver disease.

## 1. Introduction

Alcoholic liver disease (ALD) is the hepatic manifestation of alcohol overconsumption. ALD could be generally aroused by excessive and acute ingestion of alcohol [1,2]. According to disease courses and severity, there are many forms of ALD, including fatty liver, alcoholic hepatitis, and chronic hepatitis with hepatic fibrosis or cirrhosis [3,4]. It has been the major cause of liver diseases in western countries. In the United States, the predicted number of ALD patients is over two million [5]. In China, according to general statistics, there are about 300 million drinkers. Varying degrees of alcoholic fatty liver can be observed in about 80% of the people in this group, and 20% have suffered from severe alcoholic liver disease [6]. With the increasing population of heavy drinkers, especially in northern regions, ALD has become the second-largest liver disease in China, following the viral hepatitis A and hepatitis B [7].

ALD has also given rise to a high mortality rate because of ineffective treatments. The Veterans Administration Cooperative Studies have shown that the mortality rate of patients with alcoholic hepatitis and cirrhosis exceeds 60% after 4 years [4,8,9]. Until now, no satisfactory effects have been obtained by the treatments for ALD. This is mainly attributed to the severe side effects of the drugs used in current treatments [10]. Thus, ALD has become a major cause of morbidity and mortality worldwide [11,12]. Various mechanisms have been proposed as the consequences of ALD, such as inflammatory cytokines, mitochondrial injury, and oxidative stress. Thus, the development of treatment for ALD has mainly focused on drugs targeting these mechanisms, aiming to find safer treatments with fewer side effects [13,14].

Food-derived oligopeptides are functional components derived from food protein and exist in dietary protein with a specific amino acid sequence. After the protein is degraded, these functional peptides are released and exhibit superior biological activities in the process, including antioxidant, anti-inflammatory, immune regulating, antibacterial, etc. Recently, several bioactive peptides extracted from natural products were studied, such as ganodermalucidum peptides [15] and cassia seed peptides [16]. The protective impacts of these peptides against hepatic damage induced by D-galactosamine or acetaminophen have been proven [4]. Thus, the ALD-protective ingredients extracted from plants and animals have attracted attention from researchers in academia and industry. Numerous studies have confirmed that food-derived oligopeptides can be used as potential dietary therapeutics and new functional raw materials for enhancing health [17].

Corn gluten meal (CGM) is a byproduct in the starch processing industry, and it includes roughly 60% (*w*/*w*) protein [18]. Corn oligopeptides (CPs) are low molecular weight peptides decomposed from CGM using enzymolysis [18,19]. In previous studies, multiple functions of CPs have been studied and verified, such as the alleviation of fatigue, resistance to the peroxidative reaction of lipids, suppression of angiotensin I-converting enzyme, and promotion of ethanolic metabolism [1,19,20]. However, few studies have detailed the effects and relevant mechanism of CPs on ALD, although existing results have shown that they would be beneficial for treating ALD due to their antioxidant activity and ethanolic metabolism promotion [3].

This study investigated the effects of CPs on alcoholic hepatic damage in an animal model of mice. Biochemical markers assessed hepatic damage and treatment effects in mice serum. In addition, the primary culture of KCs and relevant cytokines were applied to explore the potential molecular mechanism of protective effects from CPs on alcoholic hepatic damage. This work aims to provide a reference for functional activity research of anti-inflammatories in ADL, nutritional food development, and the clinical applications of CPs.

## 2. Materials and Methods

### 2.1. Preparation of CPs

CPs were prepared from CGM [21]. The CPs involved in this study were provided by CF Haishi Biotechnology Ltd. Co. (Beijing, China). The CPs were prepared with the following protocols: the CGM was suspended in distilled water after being ground with a 60-mesh sieve (1:10, *w*/*w*). The suspension was then hydrolyzed at pH 11.0 and 90 °C for 1 h. The suspension was neutralized and centrifuged to recover the insoluble protein precipitate. The insoluble protein precipitate was resuspended and subjected to the procedures above. The wet corn protein isolate (CPI) was thus obtained.

The wet CPI was resuspended with a concentration of 6% (*w*/*w*) and subjected to a two-step enzymatic hydrolysis. In the first step, the enzymatic hydrolysis was performed with crude alkaline proteinases at pH 8.5 and 55 °C for 3 h. The second step was performed with crude neutral proteinases at pH 7.0 and 45 °C for 2 h (Angel Yeast, Hubei, China). The obtained hydrolysates were centrifuged to remove the insoluble impurities. The supernatant was filtered successively through 10 and 1 kDa MWCO ceramic membranes. The step of nanofiltration was carried out to remove the mineral salt. The salt-removed solution was concentrated by cry concentration under vacuum at 70 ℃, with an evaporation rate of 500 kg/h. When the solution concentration was about 30 Baumé degrees, it was decolored with 12% active carbon at 75 °C for 1 h. The carbon was then removed by normal filtration after de-coloration. Most of the water was removed by spray drying with a pressure of 20 MPa. The CP powder was obtained, and it was applied in the following experiments.

### 2.2. Identification of Corn Oligopeptides

CPs have been analyzed to determine the chemical components, amino acid compositions, and molecular weight distribution. The crude protein, moisture, and ash content were determined according to the methods specified by the Association of Official Analytical Chemists. The amino acid composition was determined using an amino acid analyzer (L-8900, Hitachi, Tokyo, Japan). The molecular weight distribution of the corn oligopeptide was established using HPLC (LC-20AD, Shimadzu, Kyoto, Japan) according to the previously reported method. A total amino acid analysis was conducted with an amino acid analyzer (835–50, Hitachi, Tokyo, Japan) [4,22]. The amino acid sequences of CPs were detected by HPLC-MS/MS (8060, Shimadzu Corporation, Japan) with reference to Wei’s method [23], and their main structures were revealed. Contents of the peptides in CPs were estimated using LC−MS/MS in the multiple reaction monitoring (MRM) mode by using the same RP-HPLC condition as described above. The synthetic peptides were used for the optimization of the MRM condition using LabSolution Ver. 5.80 software.

### 2.3. Animal Models

Male Kunming mice were used for this study (approval number: 2016-0006). The mice were aged 6 to 12 weeks and weighed 20 ± 2 g. All the mice were maintained in an environmentally controlled room at 22 ± 1 °C, with a 12 h light/dark cycle (light from 7:00 to 19:00). The treatment and maintenance of animals were conducted according to the Principle of Laboratory Animal Care (NIH Publication No. 85-23, revised 1985) and the Peking University Animal Research Committee guidelines.

All mice were fed a normal AIN-93M rodent diet (Vital River Limited Company, Beijing, China), and the main protein source was casein. The animals were randomly assigned to 5 groups: a normal control group (defined as the standard group; *n* = 5), an alcohol control group (defined as the control group; *n* = 10), and 3 CPs intervention groups with different doses (designated as the LCP, MCP, and HCP groups; *n* = 10).

The mice in the control and experimental groups were administered with 50% ethanol on day 7 in addition to the standard diet with a dose of 12 mL/kg of body weight. The standard group was administered with saline solution in the same manner. In the experimental groups, mice were pretreated with CPs 1 h before the ethanol administration, and the dose was 0.2, 0.4, and 0.8 g/kg of body weight (respectively designated as CP-0.2, CP-0.4, and CP-0.8) (the amount of corn peptides to be gavaged was based on the optimal recommended daily dosage for humans (≤4.5 g/day)). The mice in the control group were administrated with ethanol, without any treatment. All the mice fasted for 12 h after ethanol treatment. Then, all the mice were anesthetized with pentobarbital. Blood was taken from the mouse’s heart, and serum was obtained from the blood by centrifugation at 3000× *g* for 20 min at room temperature. The liver was carefully removed and immediately frozen in liquid nitrogen and stored in a −80° freezer until use.

### 2.4. Evaluation of Hepatic Biomarkers

#### 2.4.1. Enzyme Activities

The activities of alanine aminotransferase (ALT) and aspartate aminotransferase (AST) in mice serum were analyzed by the Mouse Aspartate Aminotransferase ELISA Kit (Cusabio Biotech Co., Ltd. Lot: CSB-E12649m, Wuhan, China) and Mouse Alanine Aminotransferase ELISA Kit (Cusabio Biotech Co., Ltd. Lot: CSB-E16539m, Wuhan, China) following the manufacturer’s instructions.

#### 2.4.2. ELISA for TNF-α, IL-1, and IL-6 in Mice Serum

Mouse serum samples were analyzed for TNF-α, IL-1, and IL-6 levels with ELISA following the manufacturer’s instructions. The involved ELISA kit was obtained from Jiancheng Bioengineering Institute (Nanjing, China).

#### 2.4.3. Real-Time Quantitative PCR

Total RNA was extracted from the liver with the SV Total RNA Isolation System. cDNA was synthesized from 1 μg of RNA by the First Strand cDNA Synthesis Kit for RT-PCR (AMV). Real-time PCR was performed for TNF-a, IL-1, IL-6, and the housekeeping gene, encoding glyceraldehyde-3-phosphate-dehydrogenase (GAPDH). It was carried out with ABI PRISM 7000 Sequence Detection systems (Applied Biosystems, USA). The reaction mixture was composed of Absolute TM QPCR SYBR Green Mixes (12.5 μL), forward and reverse primers (5 μM and 1 μL each), nuclease-free water (8 μL), and a cDNA sample (2.5 μL).

All primers were synthesized by Invitrogen (Invitrogen, Hong Kong, China). The GAPDH gene was used as an internal control. The real-time PCR primer sequences for these genes are shown in Table 1. The PCR conditions were as follows: 30 s at 95 °C for 1 cycle; 5 s at 95 °C; 31 s at 60 °C for 45 cycles; 15 s at 95 °C; 1 min at 60 °C; and 15 s at 95 °C. Results were analyzed with ABI sequence Detection System software (Applied Biosystems, Foster, CA, USA).

#### 2.4.4. Isolation and Culture of Murine Kupffer Cells (KCs)

KCs were isolated according to the method described previously [24]. Briefly, each liver was first perfused with calcium- and magnesium-free D-Hank’s solution until the liver became completely blanched. After gently mashing the liver, the digestion was allowed to proceed in this solution for 10 min at 37 °C, with stirring. It was then centrifuged at 300× *g* at 4 °C for 5 min, and the pellet was washed three times with 40 mL cold HBSS containing 10 mmol/L HEPES and 10 mg/mL DNase (HBSS + HEPES/DNase). The final pellet was resuspended in 40 mL HBSS + HEPES/DNase and centrifuged at 100× *g* at 4 °C for 1 min. The resulting supernatant (containing most of the hepatic non-parenchymal cells) was layered on a sterile Percoll gradient (15 mL 25% Percoll over 15 mL 50% Percoll), which was then centrifuged at 900× *g* for 20 min. The lower zone, including the interface zone, was collected and resuspended in 40 mL cold HBSS + HEPES/DNase and centrifuged at 900× *g* for 5 min. The pellet was resuspended, washed twice with HBSS + HEPES/DNase, and then resuspended in RPMI 1640 medium with 10 mmol/L HEPES. The final cell pellets were resuspended in the appropriate volume of RPMI 1640 medium supplemented with 10% endotoxin-free FCS, 100 U/mL penicillin, 100 U/mL streptomycins, 15 mM L-glutamine, and 10 mM HEPES to achieve a final concentration of 1 × 10^6^ viable cells/mL. The cell suspensions were seeded in 90 mm culture dishes and incubated at 37 °C in 5% CO_2_ air for 3 h to allow the adhesion of KCs. Non-adherent cells were removed by vigorous washing with Hank’s solution. Over 90% of the adherent cells were identified to be KCs by positive peroxide staining. The adherent cells were then trypsinized for cell detachment. They were sub-cultured at 2 × 10^6^ cells/mL in 35 mm dishes for 24 h to recover from isolation and adherence for in vitro stimulation. The cells were exposed to *E. coli* LPS (60 ng/mL); at the same time, the CPs with a concentration of 0.1, 0.5, and 1 mg/mL were added respectively and incubated at 37 °C for 1 h or 12 h. Kupffer cell viability with CPs was assessed using the MTT, as described previously [25].

#### 2.4.5. Extraction of Proteins

After 1 h incubation, the cells (2 × 10^6^) were washed with ice-cold phosphate-buffered saline (PBS) and were lysed by adding ice-cold SDS sample buffer containing 62.5 mM Tris-HCL (pH 6.8), 2% SDS, 10% glycerol, 50 mM DTT, and 0.1% bromphenol blue. The cell extract was collected into a microfuge tube and was sonicated for 10 to 15 s to shear DNA and reduce sample viscosity. The sample was followed by heating to 95 °C to 100 °C for 5 min, and it was spun for 20 min at 4 °C at 15,000 g in a microfuge. The supernatant was decanted, and a small aliquot was removed for protein assessment. The rest of the sample was aliquoted and frozen at −70 °C until the application for western blotting.

### 2.5. Western Blot Analyses

Protein samples were resolved by SDS-PAGE and transferred to nitrocellulose membranes (Roche Diagnostics, Rotkreuz, Switzerland). These nitrocellulose membranes were blocked with 5% skimmed milk powder/Tris-buffered saline with 0.1% Tween20 (*w*/*v*) at 4 °C overnight. After that, the membranes were incubated with the primary antibodies:inhibitor of κB kinase-α (IκB-α; Santa Cruz Biotechnology, Delaware Ave Santa Cruz, CA, USA; dilution ratio; 1:1000), TLR4, CD14 (Santa Cruz Biotechnology, USA; dilution ratio; 1:200), p-p38, p-JNK, p-ERK, total p38, total JNK, total ERK (Cell Signaling Technology, Boston, CA, USA; all at 1:200), and β-actin (Cell Signaling Technology, Boston, CA, USA; 1:500); the samples were incubated overnight at 4 °C. Then, membranes were treated with HRP-conjugated anti-rabbit IgG (H + L) as the second antibody (Promega, Madison, WI, USA; dilution ratio; 1:2000). Chemiluminescent HRP Substrate examined the immunoblotting (Cat. NO: WBKLS0100; ImmobilonTMWestern, MA, USA) according to the manufacturer’s instructions. Membranes were exposed by a FUJIFILM Luminescent Image Analyzer LAS-1000 (Macintosh TM, USA). The intensities of the resulting bands were quantified by Quantity One software on an AGS-800 densitometer (BioRad, Hercules, CA, USA).

### 2.6. Statistical Analyses

The data were expressed as mean ± SD. ANOVA and multiple comparisons were applied to calculate the statistical difference between groups. The significance level was set at 95%. All statistical calculations were performed with the SPSS 21.0 software for windows.

## 3. Results

### 3.1. The Chemical Composition of CPs

Table 2 displays the specific chemical compositions obtained by the analysis and indicates that corn oligopeptide had a high protein (83.60%) and peptide (79.23%) content. Other components in the powder had 3.78% water and 3.90% ash. The results of Table 1 show that corn oligopeptide contained considerable amounts of branched-chain amino acids (21.44%), including valine (2.72%), leucine (16.73%), and isoleucine (1.99%). Rich branched-chain amino acid compositions play an essential role in the regulation of human liver physiological functions.

Table 3 and Figure 1 show that 96.51% of the peptides in the CPs were below a molecular weight of 1000 Da and that the average molecular weight in the CP mixture was 349 Da. The average molecular weight of amino acids was 137 Da, and the mean peptide length was approximately 2.5 residues. Peptide compositions of CPs mostly were represented by dipeptides or tripeptides, which can be absorbed and transported more efficiently than either amino acids or intact proteins.

Figure 2 shows that the main peptide sequences in the CPs are pEL, LL, VL, and TL, and their proportions in the CPs are 0.72%, 0.27%, 0.17%, and 0.14%, respectively. Previous studies have shown that VL and LL exhibit protect liver effects [1,26] and TL exhibits inflammatory level inhibitory effects [27,28]. Therefore, CPS may have anti-inflammatory and protect liver functions due to the presence of these peptides.

### 3.2. The Inhibition Effects of CPs on AST and ALT in Mice Serum

AST and ALT exist in organs, including the liver, heart, skeletal muscle, and kidneys. The concentrations of AST and ALT in serum were maintained within a certain level. When there was liver cell damage, the levels of AST and ALT increased correspondingly. Therefore, in clinical practice, liver damage will be reflected by testing AST and ALT. In this work, the ALT and AST in the standard group were less than 70 IU/L and 100 IU/L (Figure 3A,B). With the treatment of ethanol, the levels of AST and ALT were increased to more than 600 IU/L and 180 IU/L. If mice were pretreated with CPs (0.2, 0.4, and 0.8 g/kg of body weight), after ethanol treatment, the levels of AST and ALT were reduced with the increased concentration of CPs (Figure 3A,B). In the group of CP-0.8, the levels of both AST and ALT were close to that of the standard group without ethanol treatment, with no significant damage to liver cells. This indicated that CPs had an excellent protective effect on the liver.

### 3.3. The Inhibition Effects of CPs on Inflammatory Factors in Mice Serum

IL-1, IL-6, and TNF-α are pro-inflammatory cytokines in the inflammatory response. The pathogenic role of pro-inflammatory factor signaling in ALD is attributable to the activation of the inflammatory response. In this work, after the treatment of ethanol, the level of IL-1, IL-6, and TNF-α were all dramatically increased compared with standard groups without ethanol (Figure 4A–C). However, with the pretreatment of CPs, the factors maintained a relatively low concentration. The level was decreased with the increased CPs concentration.

The mRNA expression levels of pro-inflammatory cytokines IL-1, IL-6, and TNF-α in the liver were analyzed after ethanol treatment with or without CPs pretreatment. Real-time PCR was performed to quantify the mRNA expression level. This is expressed as the ratio to the housekeeping gene encoding GAPDH. In this study, the mRNA expression levels of IL-1, IL-6, and TNF-α were upregulated after ethanol treatment, compared to that of the standard group without any treatment (Figure 5A–C). With the pretreatment of CPs (CP-0.8), the IL-1, IL-6, and TNF-α were maintained at relatively low levels compared to the control group after ethanol treatment (Figure 5A–C). These results indicated that CPs played an essential role in regulating the mRNA expression levels of inflammatory cytokines.

### 3.4. The Molecular Mechanism of Effects from CPs on LPS-Induced Kuffer Cells (KCs)

The LPS-induced signaling is typical and critical in the initiation and process of ALD. LPS can be recognized by cell differentiation antigen (CD14) and toll-like receptor 4 (TLR4). The downstream signaling pathways are activated and end in activating transcription factors, including nuclear factor (NF)-κB [13,29].

The cell viability of Kupffer cells was determined after treatment with different concentrations of CPs (0.1, 0.5, 1, 2, 5, 10 mg/mL). The results are shown in Figure 6. Compared with the control, the cell viability after treatment with 1 mg/mL did not show significant statistical difference for 12 h, while the treatment at the concentration of 2–10 mg/mL slightly decreased the cell viability; thus, the optimal concentration of CPS is below 2 mg/mL.

The western blot analysis was used to compare the expression of CD14 and TLR4 in three groups: standard group without any treatment, LPS treated group, and both LPS and CPs treated group (1 mg/mL, Figure 7A). In the LPS treated group, the expression level of CD14 was upregulated to be 150% and 180%, respectively, compared to the expression of β-actin (Figure 7B,C). However, in the presence of CPs with a concentration of 1 mg/mL, the expression of CD14 and TLR4 in the LPS treated group was down-regulated. The expression level was approximately equal to or a little more than that in the standard group (Figure 7B,C).

In the LPS treated group, the NF-κB was activated, and JNK, ERK, and p-38 protein phosphorylation was enhanced. This led to an inflammatory response and the upregulated expression of inflammatory cytokines, such as TNF-α and IL-1. It also led to additional liver cell damage. In the western blot analysis, after adding both CPs and LPS, the expression level of IkB-α was increased, and its inhibition effect on NF-κB was enhanced (Figure 2D). As a result, the phosphorylation of JNK, ERK, and p-38 protein was down-regulated compared to the LPS treated control group (Figure 7E–G). After adding CPs and LPS, the expression of TNF-α and IL-1 were significantly down-regulated, although the level was still higher than that in the standard group and the group treated only with CPs (Figure 7H,I). Therefore, CPs as a functional food have the potential activity to prevent up-regulation of inflammation levels induced by alcohol.

## 4. Discussion

The pathogenesis of ALD involves many factors, such as genetics and nutrition, in addition to many injurious factors such as oxidative stress, bacterial lipopolysaccharides (LPS) and cytokines [30,31]. The development of treatment of ALD has been limited since the 1970s [11,32], and most of the treatments have been associated with side effects. As a result, more and more research is beginning to explore new treatments that combine the pathogenesis of ALD from natural active ingredients with the ability to protect the liver from alcohol damage.

Maize is a popular food for people all over the world. Studies have proven the safety of CPs. They also have various functions, such as antioxidant, anti-inflammatory, anti-hypertensive, immune boosting, and anti-fatigue [4,17]. Some researchers have found a protective effect of CPs against early alcoholic liver injury in mice and rats [33,34]. They focused mainly on AST, ALT, and SOD activity and MDA levels in serum. In addition, abnormal lipid metabolism could be improved [4]. Our results found that maize peptides contain a large number of branched-chain amino acids, including valine, leucine, and isoleucine. The study showed that oral BCAA supplements improved the manifestations of recurrent hepatic encephalopathy in patients with cirrhosis, without effects on mortality, nutrition, or adverse events [35]. Tedesco, Laura et al. found that branched-chain amino acid supplementation could be used to reduce the risk of cirrhosis, improves mitochondrial functional integrity against EtOH toxicity, and preserves liver integrity in mammals [36].

In recent years, an increasing number of studies have focused on the inflammatory mechanisms (innate immune mechanisms) of ALD. It is believed that activation of KCs is a major trigger of hepatotoxicity and liver injury [37]. KCs are macrophages in the liver and account for 15% of all liver cells. lPS is phagocytosed by KCs, which then produce TNF-α and other inflammatory mediators. Degeneration and necrosis of hepatocytes is promoted [38,39]. The LPS-Kupffer cell signaling modality has been implicated in the protective effect of CPs against ALD. Previous studies have demonstrated that CPs are an effective model for screening drugs for ALD treatment [40,41]. The researchers demonstrated that the protective effect of the Chinese herbal formula Qing Gan Hou Pu Fang was achieved by regulating related molecules in a similar signaling pathway to LPS-KCs.

In previous studies, ethanol ingestion led to damage to the gut barrier. Gut barrier dysfunction results in an elevation of circling bacterial endotoxins, which plays a key role in triggering LPS as recognized by the ethanol-induced expression of the CD14/TLR4 receptor. This leads to the production of pro-inflammatory cytokines. In our study, we found that liver tissue inflammatory factors were abundantly expressed. We speculated that this was caused by elevated LPS [42]. Therefore, we examined the role of LPS in inducing inflammation onset in KCs cells. The results showed that the presence of CPs tended to reduce the expression of CD14/TLR4 on the cell membrane of KCs. The translocation of NF-κB was significantly inhibited in the nucleus of KCs. Then, the phosphorylation of JNK, ERK, and p-38 protein was down-regulated. The expression of inflammatory cytokines such as TNF-α and IL-1 could be reduced. Thus, the inflammatory response could be inhibited. The results were consistent with the concentration variation of TNF-α and IL-1 in serum and the level of ALT and AST. Therefore, it was concluded that CPs could inhibit intracellular inflammation by improving LPS-induced KCs cell activation and reducing intracellular NF-κB and MAPK cell pathway activation to protect mice from hepatocyte injury. However, it is worth noting that, although the CPs have natural ingredients, more clinical studies should be carried out to prove the clinical performance of CPs on ALD. KCs signal pathway-based evaluation methods could also be applied to evaluate other natural ingredients in ALD treatments.

## 5. Conclusions

In this study, the role of natural CP in protecting the liver from alcohol damage by inhibiting LPS-induced activation of KCs cells was successfully demonstrated. It proves to be potential food therapy for patients with ALD. The primary culture was involved in evaluating the effect mechanism from CPs on KCs. The KCs-related signal pathway could also be applied to evaluate other treatments for ALD. In addition, the KCs are closely related to liver transplants. Further studies should be performed to develop treatments based on the functions of CPs and other natural ingredients.

## Figures and Tables

**Figure 1 nutrients-14-04194-f001:**
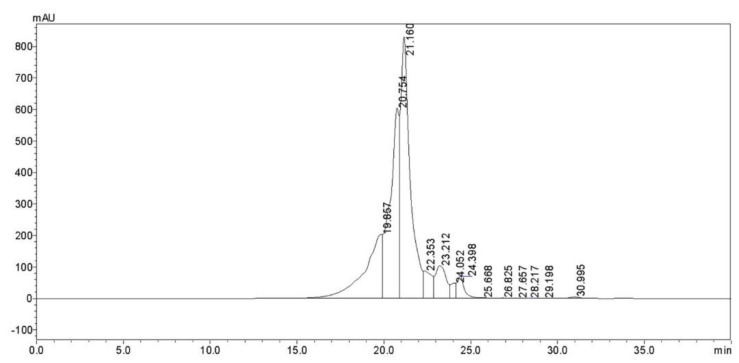
Corn oligopeptide chromatogram (220 nm).

**Figure 2 nutrients-14-04194-f002:**
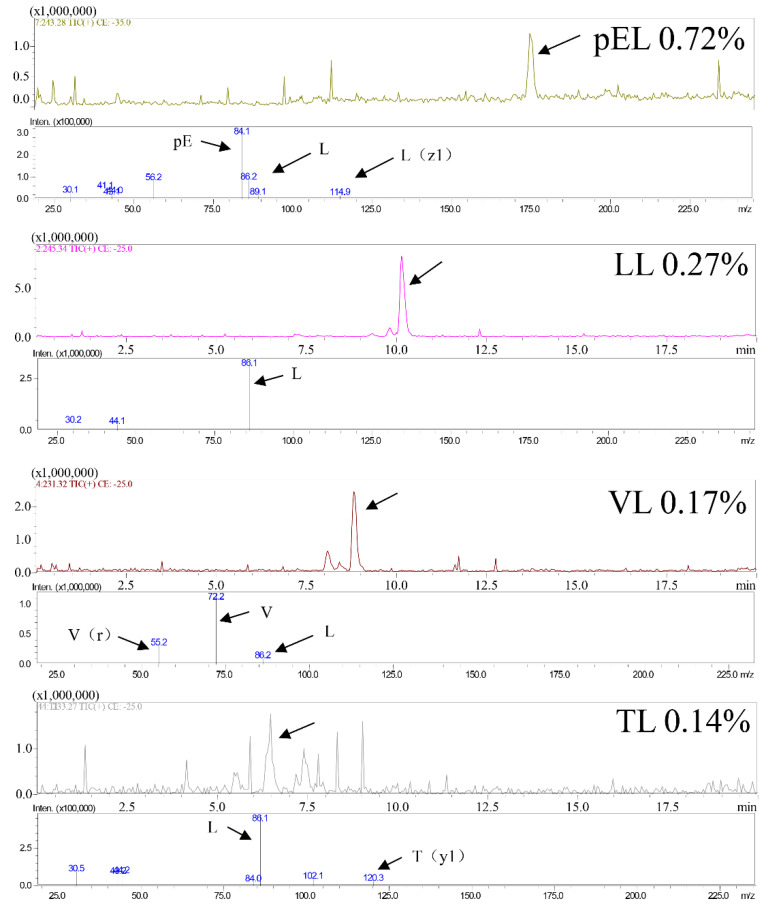
CPs main peptides sequences.

**Figure 3 nutrients-14-04194-f003:**
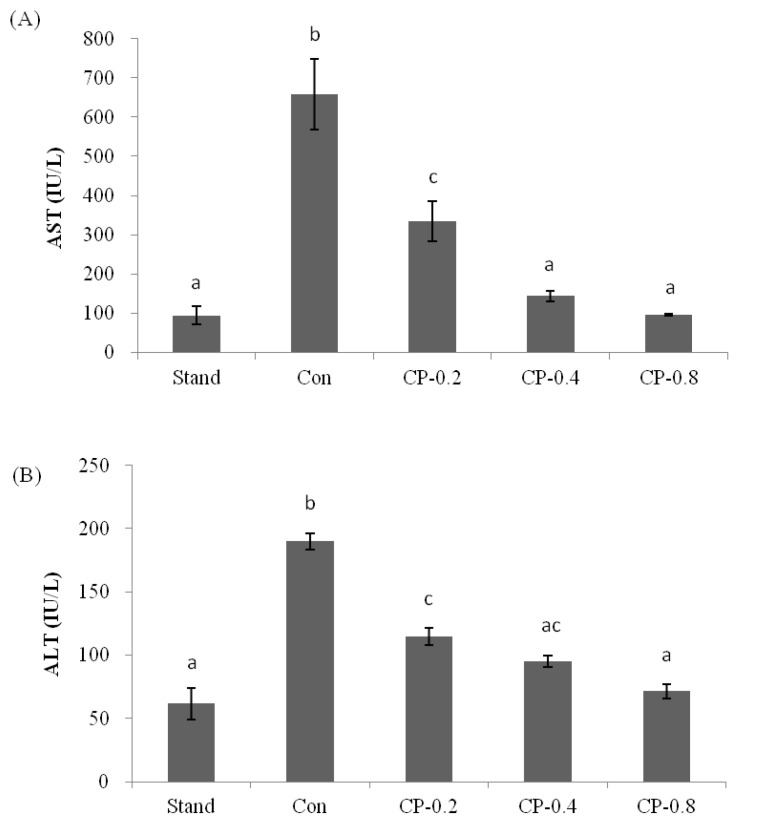
The effects of CPs on transaminase activities in mice serum at 12 h after ethanol treatment. (**A**) The level of AST. (**B**) The level of ALT. Values are expressed as means ± SEM (*n* = 5–10). Values with different letters are significantly different (*p* < 0.05).

**Figure 4 nutrients-14-04194-f004:**
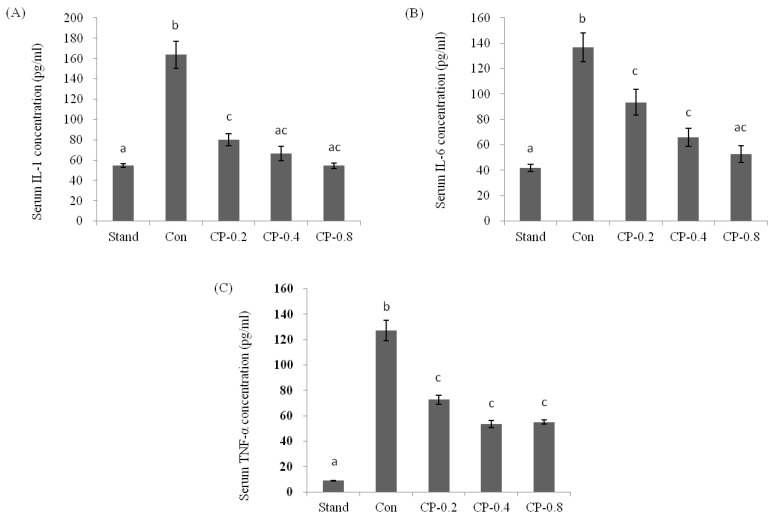
The effects of CPs on the levels of IL-1 (**A**), IL-6 (**B**), and TNF-α (**C**) in mice serum at 12 h after ethanol treatment. Values are expressed as mean ± SEM (*n* = 5–10). Values with different letters are significantly different at *p* < 0.05.

**Figure 5 nutrients-14-04194-f005:**
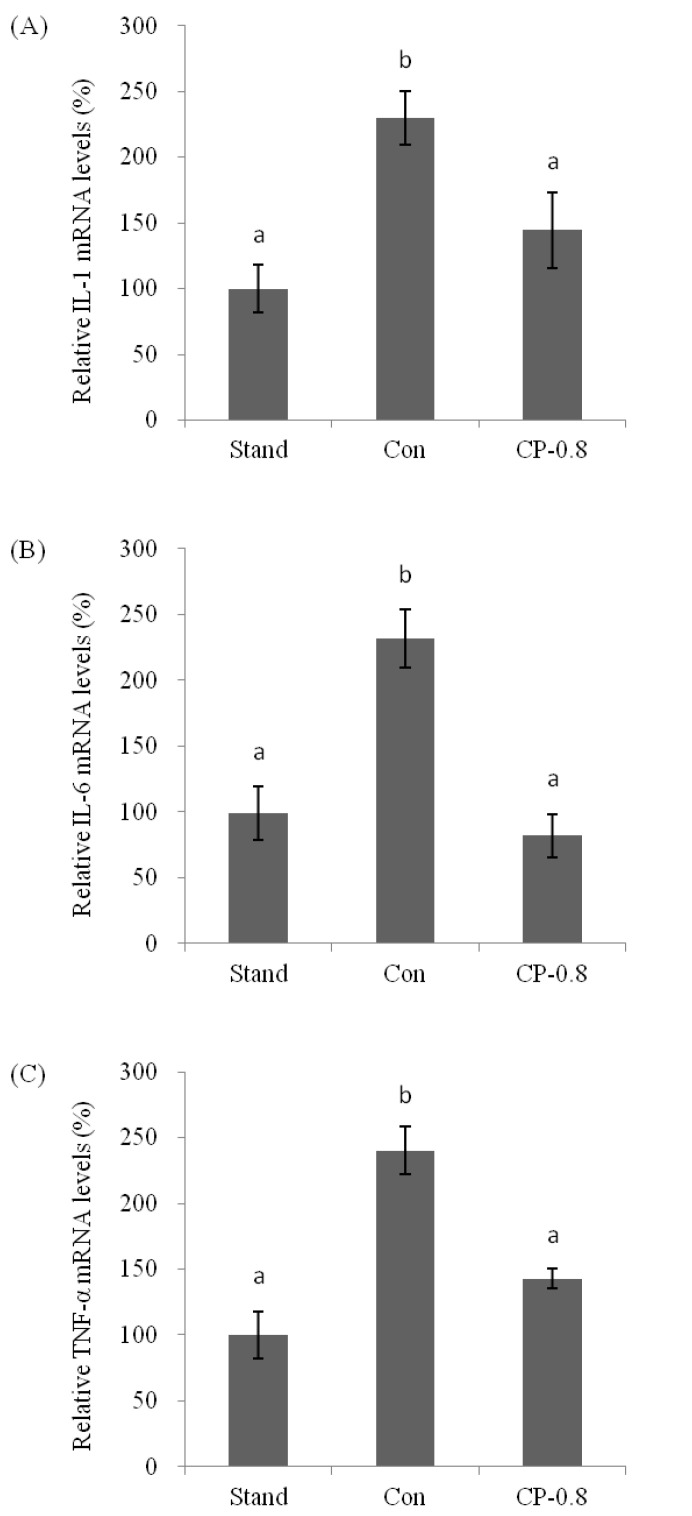
The effects of CPs on the mRNA expression of IL-1 (**A**), IL-6 (**B**), and TNF-α (**C**) in mice liver at 12 h after ethanol treatment. Values are expressed as means ± SEM (*n* = 5–10). Values with different letters are significantly different at *p* < 0.05.

**Figure 6 nutrients-14-04194-f006:**
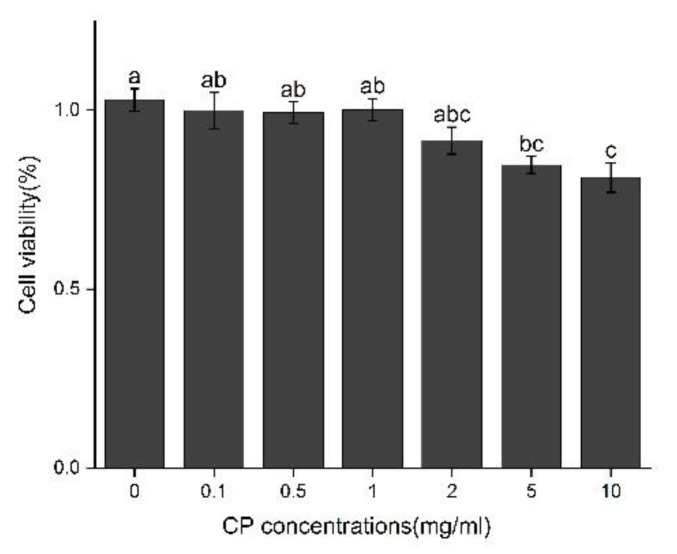
Effect of CPs concentration on cell viability in Kupffer cells after 12 h treatment. Different letters represented the significant difference at *p* < 0.05.

**Figure 7 nutrients-14-04194-f007:**
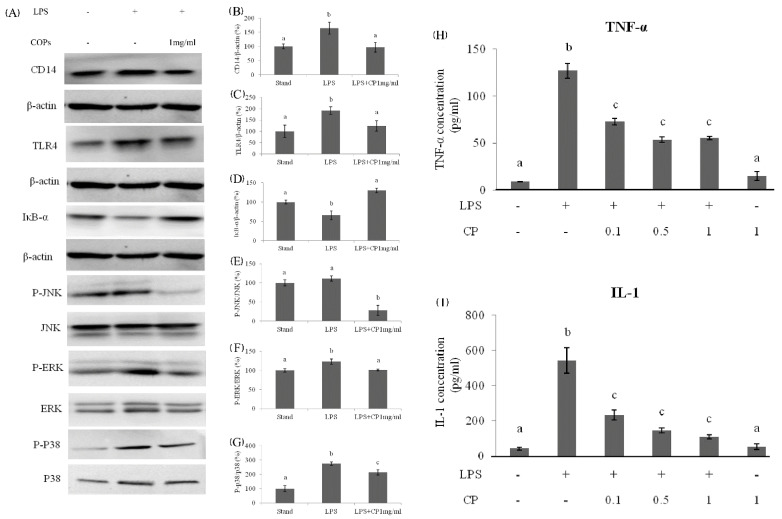
The effects of protein expression at 1 h after LPS administration in Kupffer cells: (**A**) Bands of proteins. The values of CD14 (**B**) and TLR4 (**C**) were normalized by the value of the total protein. (**D**) The effects of CPs on IkB-α degradation at 1 h after LPS administration in Kupffer cells. The expression levels of phosphorylated and total JNK, ERK, and p-38 protein were detected by western blotting. The value of JNK (**E**), ERK (**F**), and p-38 (**G**) were normalized by the value of the total protein. The effects of CPs on the TNF-α (**H**) and IL-1 (**I**) expression level in supernatants of KCs at 12 h after LPS treatment. Different letters represented the significant difference at *p* < 0.05.

**Table 1 nutrients-14-04194-t001:** PCR primers of mice used in this study.

Gene	Primer Sequences (5′-3′)	Product Length
TNF-α	forward: CATCTTCTCAAAATTCGAGTGACAA	447 bp
	reverse: TGGGAGTAGACAAGGTACAACCC	
IL-1	forward: CTTCATCTTTGAAGAAGAGCCC	418 bp
	reverse: CTCTGCAGACTCAAACTCCAC	
IL-6	forward: TTCACAAGTCCGGACAGGAG	488 bp
	reverse: TGGTCTTGGTCCTTAGCCAC 3	
GAPDH	forward: GAAGGTGAAGGTCGGAGTCA	402 bp
	reverse: TTCACACCCATGACGAACAT	

**Table 2 nutrients-14-04194-t002:** Chemical composition of CPs.

	Corn Oligopetides
Moisture (%)	3.78 ± 0.11
Ash (%)	3.90 ± 0.13
Protein (%)	83.60 ± 1.21
Peptide (%)	79.23 ± 1.19
**Amino acid composition (%)**	
Ala	8.17
Pro	6.52
Val	2.72
Met	1.97
Ile	1.99
Leu	16.73
Phe	4.33
Trp	0.23
Asp	4.73
Ser	4.14
Glu	22.72
His	1.03
Gly	1.18
Arg	1.35
Thr	2.22
Cys	2.17
Tyr	4.29
Lys	0.25

**Table 3 nutrients-14-04194-t003:** Molecular weight distributions of corn oligopeptides.

Molecular Weight (u)	Over 10,000	3000–10,000	1000–3000	150–1000	Below 150
Distributions (%)	0.0000	0.1513	3.3431	77.5960	18.9096

## Data Availability

Not applicated.

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
