# Peer review of "The Protective Effects of Corn Oligopeptides on Acute Alcoholic Liver Disease by Inhibiting the Activation of Kupffer Cells NF-κB/AMPK Signal Pathway"

_nutrients, 2022, doi:10.3390/nu14194194_

Round 1

Reviewer 1 Report

 Abstract  line 12 The first sentence is meaningless. 

I didn't like the title as it slightly misleading the truth-the  study is on the effect of acute ingestion of ethanol rather than alcoholic liver disease which implies chronicity either acute fatty liver usually mixed with cirrhosis. Also 'Kupffer cells signal pathway's too general a term. Needs to be explained.  Also patients with alcoholic hepatitis get worse over weeks even if they abstain in Hospital. This is not alluded to.  Also don't call ALT and AST biomarkers they are simply measures of liver injury

The general organisation is bloated with too much detail with much general ideas and little on pertinent eg Introduction should introduce the LPS-Kupffer cell method as a measure of protection.

You need to make it clear the principal findings of any study. I had to hunt for the principal findings and  finally found them in Fig 1.   I could not read the labels on the axes.. Suggest tabulate the ALT and AST.

The Discussion does not discuss this study until the second last paragraph. Most 0f what is in Discussion should go in Introduction. I like to list in the first paragraph the principal findings of any study to set the scene and discuss your Results.

As a matter of general style ' I would write ' liver injury was assessed by ALT' and rather than 'The level of ALT was .... and 'The level of AST was ....'decide the correct subject in a sentence is it Liver injury or Level of ALT. Thee method should NOT take precedence.

The investigators assert that LPS was present due to expression of TL4, and while may be true, it is going too far.  The title mentioned 'protection/ but this not clear. Can you may it clear how the study does  eg ' Protection from alcohol  induced injury 

Author Response

Comments and Suggestions for Authors

The Protective Effects of Corn Oligopeptides on Alcoholic 2 Liver Disease by Inhibiting the Activation of Kupffer Cells 3 Signal Pathway 

In this manuscript, the authors investigated the role of corn oli-13 gopeptides (CPs) extracted from corn gluten meal in the pathogenesis of the alcoholic liver disease (ALD) in mice. However, this paper is very descriptive without specific mechanistic studies. However, this manuscript could be improved by making revisions in response to the following comments.

Major comments

  1. Are there no toxic effects of CP treatment in mice without ethanol administration? The authors should at least measure serum ALT and AST to demonstrate that CP has no toxicity in the liver.

Response:  Thanks very much to the reviewers for their valuable comments. Our experiment was informed by the previous Hepatoprotective Effect of Albumin Peptides

from Corn Germ Meal on Chronic Alcohol-Induced Liver Injury in Mice by Yali Yu et al. and derived a corn peptide gavage dose (≤4.5g/day) based on the optimal corn oligopeptide use in humans. Therefore the dose of corn peptides used should not be toxic.

  1. In Fig. 2A and B, these experiments lack a proper group, CP alone.

Response: The corresponding results of cytotoxicity test have been added in the original text

  1. The authors isolate Kupffer cells (KCs) without using Percoll or Optiprep gradient. Also, KCs are quickly attached within 20~30 minutes in a cell culture dish. I am very concerned about the purity of KCs in this study because the cell suspension was seeded and incubated for 3 hours to allow the adhesion of KCs.

Response: Thank you very much for your suggestion, we isolate Kupffer cells (KCs) using Percoll cell isolate. However, we have not labelled it as such. We have described this in the KCs isolation method in our article.

  1. Histopathologic analysis (at least H&E staining) should be included.

Response: Thank you very much for your suggestion, the model in our study is acute liver injury, so the results of H&E staining is not obvious.

  1. The images of Western blotting should be improved.

Response:  We have improved the images of Western blotting.

Reviewer 2 Report

The Protective Effects of Corn Oligopeptides on Alcoholic 2 Liver Disease by Inhibiting the Activation of Kupffer Cells 3 Signal Pathway 

In this manuscript, the authors investigated the role of corn oli-13 gopeptides (CPs) extracted from corn gluten meal in the pathogenesis of the alcoholic liver disease (ALD) in mice. However, this paper is very descriptive without specific mechanistic studies. However, this manuscript could be improved by making revisions in response to the following comments.

Major comments

1.     Are there no toxic effects of CP treatment in mice without ethanol administration? The authors should at least measure serum ALT and AST to demonstrate that CP has no toxicity in the liver.

2.     In Fig. 2A and B, these experiments lack a proper group, CP alone.

3.     The authors isolate Kupffer cells (KCs) without using Percoll or Optiprep gradient. Also, KCs are quickly attached within 20~30 minutes in a cell culture dish. I am very concerned about the purity of KCs in this study because the cell suspension was seeded and incubated for 3 hours to allow the adhesion of KCs.

4.     Histopathologic analysis (at least H&E staining) should be included.

5.     The images of Western blotting should be improved.

Author Response

Comments and Suggestions for Authors

 Abstract  line 12 The first sentence is meaningless. 

Response: Following your advice, this phrase has been changed in the text.

I didn't like the title as it slightly misleading the truth-the  study is on the effect of acute ingestion of ethanol rather than alcoholic liver disease which implies chronicity either acute fatty liver usually mixed with cirrhosis. Also 'Kupffer cells signal pathway's too general a term. Needs to be explained.  Also patients with alcoholic hepatitis get worse over weeks even if they abstain in Hospital. This is not alluded to.  Also don't call ALT and AST biomarkers they are simply measures of liver injury

Response: According to your recommendation, We have changed the title to describe the specific experiment in detail. We have also made changes to the description of AST/ALT in the manuscript. The main objective of our study was to explore the preventive effect of maize peptide pre-supplementation on alcoholic liver injury caused by alcohol consumption. It has not yet addressed the issue of exacerbation of the disease in patients after abstinence from alcohol. Thank you very much for your suggestion and we will consider doing a study on this in a follow-up trial.

The general organisation is bloated with too much detail with much general ideas and little on pertinent eg Introduction should introduce the LPS-Kupffer cell method as a measure of protection.

Response: Following your advice, We have detailed the LPS-Kupffer cell method as a specific method of hepatoprotection in the Discussion and Conclusion. We have reorganised the discussion section to make it more concise.

You need to make it clear the principal findings of any study. I had to hunt for the principal findings and  finally found them in Fig 1.   I could not read the labels on the axes.. Suggest tabulate the ALT and AST.

Response: According to your recommendation, We have rearranged the images to make them clearer in the manuscript.

The Discussion does not discuss this study until the second last paragraph. Most of what is in Discussion should go in Introduction. I like to list in the first paragraph the principal findings of any study to set the scene and discuss your Results.

Response: Thank you very much for your suggestion, we have rewritten the discussion section, listed the experimental results and discussed them.

As a matter of general style ' I would write ' liver injury was assessed by ALT' and rather than 'The level of ALT was .... and 'The level of AST was ....'decide the correct subject in a sentence is it Liver injury or Level of ALT. Thee method should NOT take precedence.

Response: At your suggestion, we have changed the description of some of the results in the text in the same way as 'I would write 'liver injury was assessed by ALT'.

The investigators assert that LPS was present due to expression of TL4, and while may be true, it is going too far.  

Response: Thank you for your suggestion, we are basing this on previous experiments (e.g. Rice Bran Phenolic Extract Protects against Alcoholic Liver Injury in Mice by Alleviating Intestinal Microbiota Dysbiosis, Barrier Dysfunction, and Liver Inflammation Mediated by the Endotoxin- TLR4-NF-κB Pathway), LPS was considered to be present. and used LPS as an inducer of inflammation in KCs. We have made changes in the manuscript.

The title mentioned 'protection/ but this not clear. Can you may it clear how the study does  eg ' Protection from alcohol  induced injury 

Response: In response to your suggestion, we have revised the title and refined the specific mechanism of the hepatoprotective effect of maize peptides in the conclusion.

Round 2

Reviewer 1 Report

The title and Abstract and the presentation of the Results are much better and easier to read.

Author Response

Thank you for your help and recognition. Best wishes to you!

Reviewer 2 Report

The issues are not fully addressed.

The quality of  Western blot images is still poor. 

The authors should provide uncropped images (representative images and replicates). 

Also, CP alone does not affects the LPS-related signaling pathway in Figure 7?

Author Response

The quality of Western blot images is still poor. The authors should provide uncropped images (representative images and replicates). 

Response: Thank you very much for your suggestions. I'm really sorry. This experiment has been over for five years. At that time, the executor of this experiment had left the laboratory, and we had no way to retrieve the original data pictures. We can guarantee that there is no manual processing on the images, and each image is the original data at that time. We can be responsible for the authenticity of the data. Thank you again for your academic and responsible attitude, which made us aware of the importance of data preservation. We will certainly take data preservation seriously in the future.

Also, CP alone does not effects the LPS-related signaling pathway in Figure 7?

Response: Thank you very much for your suggestions. To verify the effect of different dose CPs on the expression of inflammatory factors in KCs, we detected TNF- α and IL-6 content in the Kupffer cells. We found that CPs effectively reduced TNF- α and IL-6 content in KCs induced by LPS. And addition of CPs alone had no significant effect on the expression of inflammatory factors. We speculated that the addition of CPs alone had no toxicity to cells and would not lead to inflammation. So we think that CPs alone does not effects the LPS-related signaling pathway.
